# Loop counting matters in SMEFT

**Gerhard Buchalla[1], Gudrun Heinrich[2], Christoph Müller-Salditt[1] and Florian Pandler[1]**

**1** Ludwig-Maximilians-Universität München, Fakultät für Physik, Arnold Sommerfeld Center for Theoretical Physics, D–80333 München, Germany
**2** Institute for Theoretical Physics, Karlsruhe Institute of Technology (KIT), D–76128 Karlsruhe, Germany

## Abstract

We show that, in addition to the counting of canonical dimensions, a counting of loop orders is necessary to fully specify the power counting of Standard Model Effective Field Theory (SMEFT). Using concrete examples, we demonstrate that considering the canonical dimensions of operators alone may lead to inconsistent results. The counting of both, canonical dimensions and loop orders, establishes a clear hierarchy of the terms in SMEFT. In practice, this serves to identify, and focus on, the potentially dominating effects in any given high-energy process in a meaningful way. Additionally, this will lead to a consistent limitation of free parameters in SMEFT applications.

 Check for updates

# 1 Introduction

The Standard Model (SM) of particle physics can be viewed as the low-energy approximation of a more fundamental theory at higher energies that is yet to be discovered. The success of the SM and the absence of new resonances in the experiments at the Large Hadron Collider (LHC) indicate a mass gap separating the new physics from the electroweak scale. The bottom-up construction of an effective field theory (EFT) based on the SM particle content and symmetries (SMEFT) is, therefore, a well-motivated and widely adopted framework for a model-independent description of new-physics effects [1–6]. Generally speaking, such effects are encoded in operators of canonical dimension higher than four in the EFT Lagrangian.

Including or omitting any specific operator in a bottom-up EFT calculation has to rely on a clear power-counting prescription. The power counting rests on (generic) assumptions about the underlying physics at shorter distances. The assumptions, and the resulting power counting, are not unique, but this is unavoidable. Necessarily, a choice has to be made to fix this power counting in a consistent way. This holds in particular for SMEFT.

Commonly SMEFT is organized in terms of the canonical dimension of operators. The coefficient of an operator of canonical dimension $d$ scales as $\Lambda^{4-d}$ with a new-physics scale $\Lambda$, leading to increasing suppression with increasing operator dimension.

We will argue that a power counting for SMEFT based on canonical dimensions alone is incomplete. It needs to be supplemented by specifying whether SM fields are weakly or strongly coupled to the new-physics sector. This assumption is effectively described by a counting of loop orders. Keeping track of loop counting not only provides a consistent treatment of higher-dimensional operators in a given process; it also leads to a systematic combination of SMEFT corrections with calculations in perturbation theory. We are not suggesting that SMEFT with a power counting based on canonical dimensions (with order one operator coefficients) is inconsistent as an EFT under perturbative renormalization. Rather, we point out that SMEFT organized in such a way would fail to match a large class of weakly coupled UV models and, in any case, would still call for a reasoning to assign a power-counting size to the coefficients.

The basic rules of power counting on which we rely are by no means new. However, their implications are not always consistently applied, and they are often not spelled out explicitly. We will review the organizing principle of SMEFT, emphasizing in particular the role of loop counting. The relevance and use of the latter will be demonstrated with concrete examples and calculations.

The fact that canonical dimensions alone do not provide the full information needed for the power counting of SMEFT is already illustrated by the Higgs-mass operator $\phi^\dagger \phi$ in the SM Lagrangian. Carrying a canonical dimension of two, it would appear, at face value, to be dominant over the remaining SM terms of dimension four, which is certainly not the case. We will see how the missing information can be provided by loop counting.

This paper is organized as follows. In Sec. 2 we introduce the toy scenario of a heavy scalar coupled to top quarks $t$ and discuss the process $e^+ e^- \to t\bar{t}$ within an EFT where the heavy scalar is integrated out. Using this top-down example, we demonstrate how a magnetic-moment type operator $m_t \bar{t} \sigma_{\mu\nu} t F^{\mu\nu}$ and a four-fermion operator $\bar{t} t \bar{t} t$ contribute at the same order in the EFT, even though the former enters the scattering amplitude at tree level, but the latter only at one loop. We show that loop counting explains and clarifies this observation, and generalize the discussion to a bottom-up EFT treatment. In Sec. 3 we address the issues highlighted in Sec. 2 within the general context of SMEFT. We review the SMEFT power counting, emphasizing the need to include the counting of loop orders, conveniently expressed using the notion of chiral dimensions $d_\chi$, in addition to canonical dimensions $d_c$. The general counting scheme is illustrated with the example of Higgs production in gluon fusion $gg \to h$, which nicely displays the combined role of canonical dimensions and loop orders in the SMEFT

expansion. In Sec. 4 we return to the topic treated with a toy model in Sec. 2, generalizing it to the more realistic case of SMEFT, analyzing $u\bar{u} \rightarrow t\bar{t}$ via gluon exchange within a decoupling Two-Higgs Doublet Model (2HDM) as the UV completion. We finally conclude in Sec. 5.

## 2   Toy model analysis of $e^+e^- \rightarrow t\bar{t}$

We consider a toy model with an electron $\psi$ of mass $m_e \approx 0$ and a heavy fermion $t$ of mass $m$, both coupled to electromagnetism. In addition to this "standard" physics, we introduce a real scalar field $S$ with mass $M$, which has renormalizable self-interactions and a Yukawa coupling to the "top-quark" $t$.[1] The Lagrangian reads

$$\mathcal{L} = \bar{\psi}(i\slashed{D} - m_e)\psi + \bar{t}(i\slashed{D} - m)t - \frac{1}{4}F_{\mu\nu}F^{\mu\nu}$$
$$+ \frac{1}{2}(\partial S)^2 - \frac{1}{2}M^2S^2 - \frac{b}{3!}S^3 - \frac{\lambda}{4!}S^4 - g\bar{t}tS, \tag{1}$$

where

$$D_\mu = \partial_\mu + ieq_f A_\mu, \qquad q_e = -1, \quad q_t = \frac{2}{3}, \qquad F_{\mu\nu} = \partial_\mu A_\nu - \partial_\nu A_\mu. \tag{2}$$

The first line of (1) is quantum electrodynamics with two different fermions. The "non-standard" physics of the scalar in the second line is assumed to be governed by a scale $M$, which is taken to be much larger than $m$ and the typical energies ($\sqrt{s} \sim$ few times $m$) accessible in experiment. We allow $b \sim M$ and take the dimensionless couplings in (1) of order unity, unless specified otherwise. The heavy scalar $S$ modifies the dynamics of the top quark and leads at energies of order $\sqrt{s}$ to "new-physics" effects, suppressed by powers of $s/M^2$.

As an example, we take the process $e^-(k_1)e^+(k_2) \rightarrow t(p_1)\bar{t}(p_2)$. To lowest order, within the model of eq. (1), the amplitude for this process arises from $s$-channel photon exchange, shown in Fig. 1 (a). It is given by

$$\mathcal{A}_{LO} = -i\frac{e^2 q_t}{q^2}\bar{v}(k_2)\gamma_\mu u(k_1)\bar{u}(p_1)\gamma^\mu v(p_2), \tag{3}$$

where $q = p_1 + p_2 = k_1 + k_2$, $s \equiv q^2$. We are interested in the leading corrections to this amplitude from the heavy sector in the second line of (1). In terms of the $t$-quark vertex function

$$\Gamma^\mu \equiv \gamma^\mu + \delta\Gamma^\mu, \tag{4}$$

the amplitude can be written as

$$\mathcal{A} \equiv \mathcal{A}_{LO} + \delta\mathcal{A} = -i\frac{e^2 q_t}{q^2}\bar{v}(k_2)\gamma_\mu u(k_1)\bar{u}(p_1)\Gamma^\mu v(p_2), \tag{5}$$

where $\delta\Gamma^\mu$ contains the effect of $S$-boson exchange on the $t\bar{t}$-photon vertex.

### 2.1   Full theory

We first determine $\delta\Gamma^\mu$ in the full theory (1) up to order $g^2$. The relevant diagram is displayed in Fig. 1 (b). Fig. 1 (c) is used to fix the necessary counterterm. With on-shell renormalization of the $t$-quark and expanding to first order in $1/M^2$, we obtain

$$\delta\Gamma^\mu = -\frac{g^2}{16\pi^2}\frac{1}{M^2}\left[\left(\frac{\ln r}{3} + \frac{4}{9} + h_1(z)\right)q^2\gamma^\mu + \left(\ln r + \frac{7}{6} + h_2(z)\right)i\sigma^{\mu\nu}q_\nu m\right]. \tag{6}$$

---

[1] A similar model has also been considered e.g. in [7].

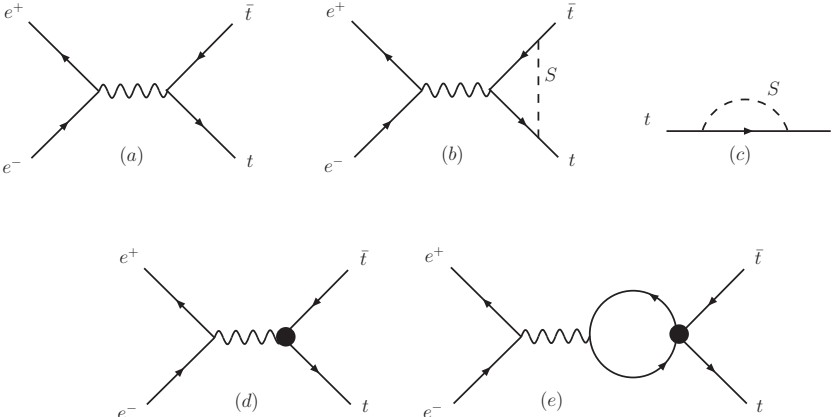

Figure 1: $e^+e^- \to t\bar{t}$ in a toy model. (a): Lowest-order amplitude. (b), (c): Leading corrections from $S$-scalar exchange (mass $M$) in the full theory. (d), (e): Contributions needed to reproduce the $1/M^2$ corrections of the full theory within the EFT. The black dots represent local operators of dimension 6. They contribute at tree level ($Q_2, Q_3$ in (d)) and at one loop ($Q_1$ in (e)). See text for further explanation.

Here we have defined

$$r = \frac{m^2}{M^2}, \qquad z = \frac{q^2}{4m^2}, \tag{7}$$

and ($\bar{x} \equiv 1 - x$)

$$h_1(z) = \int_0^1 dx \, 2x\bar{x} \ln(1 - 4x\bar{x}z - i\eta) = \left(\frac{1}{3} + \frac{1}{6z}\right) h_2(z) + \frac{1}{9}, \tag{8}$$

$$h_2(z) = \int_0^1 dx \, \ln(1 - 4x\bar{x}z - i\eta) = -2 + \sqrt{1 - \frac{1}{z}} \ln \frac{\sqrt{1 - \frac{1}{z}} + 1}{\sqrt{1 - \frac{1}{z}} - 1}. \tag{9}$$

The second expression for $h_2(z)$ is immediately applicable in the Euclidean region $z < 0$. For $z > 0$ it holds with the prescription $z \to z + i\eta$.

## 2.2 Top-down EFT

The result for $\delta\Gamma^\mu$ in (6) can be reproduced within a low-energy effective field theory of the heavy sector. The EFT takes the form of the first line in (1), supplemented by local operators of dimension 6,

$$\Delta\mathcal{L}_6 = \frac{1}{M^2} \sum_i C_i Q_i, \tag{10}$$

when we neglect higher orders in $1/M^2$. Here we assume that the heavy sector is known. The EFT can then be constructed by explicitly integrating out the scalar $S$. This scenario is commonly refered to as a top-down EFT. The relevant operators are given by

$$Q_1 = \bar{t}t\,\bar{t}t, \qquad Q_2 = \partial_\mu F^{\mu\nu}\,\bar{t}\gamma_\nu t, \qquad Q_3 = m\bar{t}\sigma_{\mu\nu}t\,F^{\mu\nu}. \tag{11}$$

$Q_1$ arises when the scalar $S$ is integrated out (removed as a propagating degree of freedom from the theory) at tree level with $C_1 = g^2/2$. $Q_2$ and $Q_3$ are generated at one loop and correspond to local terms in (6). The coefficients $C_i$ are found by matching the full-theory result for $\delta\Gamma^\mu$ to its EFT counterpart. The matching condition reads

$$-ieq_t\delta\Gamma^\mu = -ieq_t\delta\Gamma^\mu_{Q_1} + \frac{C_2}{M^2}(-i)q^2\gamma^\mu + \frac{C_3}{M^2}(-2)\sigma^{\mu\nu}q_\nu m\,, \tag{12}$$

equating the full-theory vertex function on the left with its EFT representation on the right. The latter consists of the one-loop contribution from $Q_1$ in Fig. 1 (e)

$$\delta\Gamma^\mu_{Q_1} = -\frac{1}{16\pi^2}\frac{2C_1}{M^2}\left[\left(\frac{1}{3}\ln\frac{m^2}{\mu^2} + h_1(z)\right)q^2\gamma^\mu + \left(\ln\frac{m^2}{\mu^2} + h_2(z)\right)i\sigma^{\mu\nu}q_\nu m\right]\,, \tag{13}$$

and the tree-level contributions from $Q_2$ and $Q_3$ in Fig. 1 (d). We have renormalized the vertex function in (13) using $\overline{\text{MS}}$ subtraction. Condition (12) then implies

$$C_1 = \frac{g^2}{2}\,, \qquad C_2 = -eq_t\frac{g^2}{16\pi^2}\left(\frac{1}{3}\ln\frac{\mu^2}{M^2} + \frac{4}{9}\right)\,, \qquad C_3 = eq_t\frac{g^2}{16\pi^2}\left(\frac{1}{2}\ln\frac{\mu^2}{M^2} + \frac{7}{12}\right)\,. \tag{14}$$

Together with (4) and (5), eqs. (12) – (14) reproduce the leading $1/M^2$ corrections to the $e^+e^- \to t\bar{t}$ amplitude within the EFT.

Let us summarize a few relevant aspects of this result.

i) As is well known, the EFT formulation achieves a factorization of large ($\sim M$) and small ($\sim m$) scales. Contributions from large scales are encoded in the Wilson coefficients (14), from small scales in the matrix elements of local operators, as seen in (13). The two regions are separated by a renormalization scale $\mu$, which cancels in the full amplitude.

ii) Within our approximation, operator $Q_1$ mixes into $Q_2$ and $Q_3$ under renormalization. The corresponding renormalization-group functions can be read off from (14):

$$\beta_i \equiv 16\pi^2\frac{dC_i}{d\ln\mu} \Rightarrow \qquad \beta_2 = -\frac{4}{3}eq_tC_1\,, \qquad \beta_3 = 2eq_tC_1\,. \tag{15}$$

The coefficients of the local operators $Q_2$, $Q_3$ also provide the one-loop counterterms necessary to renormalize the UV divergences originally contained in (13).

iii) Using the equations of motion, operator $Q_2$ may be eliminated in favour of the 4-fermion operator

$$Q_2' = -e\bar{\psi}\gamma^\nu\psi\,\bar{t}\gamma_\nu t + eq_t\bar{t}\gamma^\nu t\,\bar{t}\gamma_\nu t\,, \tag{16}$$

which gives an equivalent contribution to the $e^+e^- \to t\bar{t}$ amplitude.

iv) The one-loop contributions from $Q_1$ in (13) are essential to reconstruct the complete $1/M^2$ corrections within the EFT, including the non-local terms expressed by the (complex) functions $h_1(z)$ and $h_2(z)$. Such terms cannot arise from the local operators $Q_2$ and $Q_3$.

v) We note that all three operators yield corrections of the same order to the amplitude, $\sim g^2/16\pi^2M^2$. This is the case even though $Q_1$ contributes only at one-loop, whereas $Q_2$ and $Q_3$ contribute at tree level, as illustrated in Fig. 1 (d) and (e). The distinction is clearly not captured by the canonical dimension of these operators, which is six in each

case. To make the difference explicit, it is instead necessary to employ chiral dimensions $d_\chi$, which count loop orders.[2] We have

$$d_\chi[C_1 Q_1] = 4, \qquad d_\chi[C_2 Q_2] = d_\chi[C_3 Q_3] = 6. \tag{17}$$

$Q_2$ and $Q_3$ enter (10) with two units of $d_\chi$, or one loop order, higher than $Q_1$. A one-loop insertion of $Q_1$ thus contributes at the same loop order as $Q_2$ and $Q_3$ at tree level.

## 2.3 Bottom-up EFT

We next imagine a scenario in which the standard physics at energy scales $\sim m$ is still described by the first line in (1), but we do not know the physics of the heavy sector, assumed to reside at $M \gg m$. To order $1/M^2$ this physics is given by an effective Lagrangian of the form (10), where the $Q_i$ represent a basis of dimension-6 operators, with coefficients $C_i$ treated as unknown parameters. This scenario is the analogue of SMEFT, applied to our toy model. In the present case, using the field content $\psi$, $t$ and $A_\mu$ and the $U(1)$ gauge symmetry, the operators $Q_i$ may be chosen as follows.

First, there are several (hermitian) 4-fermion operators, which can be written as

$$Q_{S1} = \bar{t}t\,\bar{\psi}\psi, \qquad Q_{S2} = i\bar{t}t\,\bar{\psi}\gamma_5\psi, \qquad Q_{S3} = i\bar{t}\gamma_5 t\,\bar{\psi}\psi, \qquad Q_{S4} = \bar{t}\gamma_5 t\,\bar{\psi}\gamma_5\psi, \tag{18}$$

$$\begin{aligned} Q_{V1} &= \bar{t}\gamma^\mu t\,\bar{\psi}\gamma_\mu\psi, & Q_{V2} &= \bar{t}\gamma^\mu t\,\bar{\psi}\gamma_\mu\gamma_5\psi, \\ Q_{V3} &= \bar{t}\gamma^\mu\gamma_5 t\,\bar{\psi}\gamma_\mu\psi, & Q_{V4} &= \bar{t}\gamma^\mu\gamma_5 t\,\bar{\psi}\gamma_\mu\gamma_5\psi, \end{aligned} \tag{19}$$

$$Q_{T1} = \bar{t}\sigma^{\mu\nu}t\,\bar{\psi}\sigma_{\mu\nu}\psi, \qquad Q_{T2} = i\bar{t}\sigma^{\mu\nu}t\,\bar{\psi}\sigma_{\mu\nu}\gamma_5\psi, \tag{20}$$

and, with $\eta = t, \psi$,

$$Q_{\eta S1} = \bar{\eta}\eta\,\bar{\eta}\eta, \qquad Q_{\eta S2} = i\bar{\eta}\eta\,\bar{\eta}\gamma_5\eta, \qquad Q_{\eta S4} = \bar{\eta}\gamma_5\eta\,\bar{\eta}\gamma_5\eta, \tag{21}$$

$$Q_{\eta V1} = \bar{\eta}\gamma^\mu\eta\,\bar{\eta}\gamma_\mu\eta, \qquad Q_{\eta V2} = \bar{\eta}\gamma^\mu\eta\,\bar{\eta}\gamma_\mu\gamma_5\eta. \tag{22}$$

Fierz identities have been used to remove redundant structures.

There are no independent operators of dimension 6 built only from $F_{\mu\nu}$ and derivatives. Finally, operators with a fermion bilinear and $F_{\mu\nu}$ are (assuming CP conservation)

$$Q_{tF} = m\bar{t}\sigma_{\mu\nu}t\,F^{\mu\nu}, \qquad Q_{\psi F} = m_e\bar{\psi}\sigma_{\mu\nu}\psi\,F^{\mu\nu}. \tag{23}$$

In the model of Sec. 2.2, only three operators from this basis were generated: $Q_1 = Q_{tS1}$, $Q_3 = Q_{tF}$ and $Q_2' = -eQ_{V1} + 2/3eQ_{tV1}$.

In the bottom-up version of the EFT we are interested in parametrizing the leading corrections (in $1/M^2$) to the $e^+e^- \to t\bar{t}$ amplitude (3). Let us use (10) and assume a power counting based just on canonical dimensions. This amounts to taking all coefficients $C_i = \mathcal{O}(1)$.[3] The dominant corrections to $A_{LO}$ would then become ($\bar{v} = \bar{v}(k_2)$, etc.)

$$\delta A = \frac{i}{M^2}\sum_i C_i\langle Q_i\rangle = \frac{i}{M^2}\bar{v}\gamma_\mu u\,\bar{u}\sigma^{\mu\nu}v\,2ie\frac{mq_\nu}{q^2}C_{tF} + \frac{i}{M^2}\bar{v}\gamma_\mu u\,\bar{u}\gamma^\mu v\,C_{V1} + \dots \tag{24}$$

---

[2]Loop orders can be conveniently counted by assigning chiral dimensions $d_\chi$ to fields and weak couplings: $d_\chi = 0$ for bosons, and $d_\chi = 1$ for each derivative, fermion bilinear and weak coupling. The total chiral dimension of a term is then related to its loop order $L$ through $d_\chi \equiv 2L + 2$, see [8] and Sec. 3.2.

[3]They may be smaller, or even zero, in reality, but they are not arbitrary. In particular, they must be small compared to $M^2/m^2$, otherwise the term $C_i Q_i/M^2 \sim Q_i/m^2$ would be of the same order as the leading, dimension-4 Lagrangian, spoiling the EFT expansion.

where $\langle Q_i \rangle$ denotes the matrix element of $Q_i$. Here we show $C_{V1}$ as representative for the terms from all the 4-fermion operators of the type $(\bar\psi \ldots \psi)(\bar t \ldots t)$ that contribute at tree level. On the other hand, 4-top-quark operators such as $Q_{tS1} = \bar t t\, \bar t t$ will contribute to $\delta A$ at one-loop order. Assuming $C_{tS1} = \mathcal{O}(1)$, this implies $\delta A_{tS1} \sim 1/16\pi^2 M^2$, which appears to be subleading with respect to the terms $\sim 1/M^2$ in (24). This would leave the latter terms as the dominant corrections in the bottom-up EFT.

However, the example in Sec. 2.2 shows that such an approximation would be inconsistent. In fact, the tree-level terms in (24) are unable to reproduce the leading corrections (6) of the model with the heavy scalar in (1), because the nonlocal terms encoded in $h_{1,2}(z)$ from the one-loop matrix element of $\bar t t\, \bar t t$ are absent.

Clearly, information on the loop counting is missing in the consideration above. This information is necessary to tell us that e.g. $Q_{tF}$ and $Q_{tS1}$ do in general contribute to $\delta A$ at the same order, in both $1/16\pi^2$ and $1/M^2$, even though $Q_{tF}$ enters at tree level and $Q_{tS1}$ at one loop. Canonical dimensions alone cannot provide this information, as discussed at the end of Sec. 2.2.

We emphasize the following points:

i) Although the model in (1) is only a specific realization of the heavy-sector physics, it serves as a strict counterexample to the validity of only counting canonical dimensions: Since the bottom-up EFT is constructed in the most general, model-independent way, it must be able to reproduce any concrete model of the physics at scale $M$, to a given order in the EFT approximation.

ii) While the scalar-exchange model in (1) is only a specific scenario, its implications are generic: Any heavy boson coupled to the top quark will, in general, have the effect of inducing 4-top interactions at tree level and $Q_{tF}$ at one loop.

iii) The systematic connection between 4-top operators and $Q_{tF}$ is independent of the coupling strength: In the example of (14), with $\bar t t\, \bar t t \equiv Q_1$ and $Q_{tF} \equiv Q_3$ the ratio of coefficients is $C_3/C_1 = \mathcal{O}(1/16\pi^2)$, independent of $g$. For weak coupling, $g \sim 1$, the coefficient of the magnetic-moment operator is loop suppressed, $C_3 = \mathcal{O}(1/16\pi^2)$. $C_3 = \mathcal{O}(1)$ is possible, but only at the price of strong coupling $g \sim 4\pi$. In this case the 4-fermion coefficient would also become strong $C_1 = \mathcal{O}(16\pi^2)$.

We can be more specific about the strong-coupling case. Here we assume that the top-quark is strongly coupled to the new physics at scale $M$. In full generality, the top-quark vertex function $\bar t \Gamma^\mu t$ from (4) can be expressed in the standard way as

$$\Gamma^\mu = \gamma^\mu G_1(q^2) + \frac{i\sigma^{\mu\nu}q_\nu}{2m} G_2(q^2), \tag{25}$$

with form factors $G_{1,2}$. To lowest order, $G_1 = 1$, $G_2 = 0$. The leading new-physics effects are described by expanding the form factors to first order in $1/M^2$. This can be accomplished within a bottom-up EFT, which we may write as

$$\Delta\mathcal{L}_6 = \frac{1}{M^2}\left[ C_1\, \bar t t\, \bar t t + C_2\, \partial_\mu F^{\mu\nu}\, \bar t \gamma_\nu t + C_3\, m \bar t \sigma_{\mu\nu} t\, F^{\mu\nu} + \ldots \right]. \tag{26}$$

Here the ellipsis denotes the remaining contributions from the 4-fermion operators in the full basis. One then finds

$$G_1(q^2) = 1 + \frac{q^2}{M^2}\left[ \frac{C_2}{eq_t} - \frac{C_1}{8\pi^2}\left( \frac{1}{3}\ln\frac{m^2}{\mu^2} + h_1(z) \right) + \ldots \right], \tag{27}$$

$$G_2(q^2) = -\frac{m^2}{M^2}\left[ \frac{4C_3}{eq_t} + \frac{C_1}{4\pi^2}\left( \ln\frac{m^2}{\mu^2} + h_2(z) \right) + \ldots \right]. \tag{28}$$

We note that $C_{2,3}$ have to come with at least a factor of $e$ that is necessarily associated with $F^{\mu\nu}$ in $Q_{2,3}$. In contrast to (14), no weak couplings are associated with $C_1$ if the top quark is strongly coupled. Hence, the chiral dimension is 2 for the first term in (26), and 4 for the second and third. $C_{2,3}$ then have a loop suppression relative to $C_1$ and all coefficients contribute at the same order in (27) and (28). A similar consideration applies to the remaining terms in (26). Four-top operators $(\bar{t}\dots t)(\bar{t}\dots t)$ contribute in analogy to $Q_1$. Finally, the physical amplitude for $e^+e^- \to t\bar{t}$ also receives contributions from operators of the type $(\bar{\psi}\dots\psi)(\bar{t}\dots t)$, in addition to the photon-exchange amplitude with the form-factor term $\bar{t}\Gamma^\mu t$ from (25), (27) and (28).

## 3 SMEFT

Any bottom-up EFT of unknown physics at short distances has to be defined by specifying

    a) its low-energy degrees of freedom (particle content),

    b) the relevant local and global symmetries,

    c) its power counting.

The power counting rests on general assumptions about the underlying dynamics, whose details are necessarily left undetermined. The assumptions concern, in particular, the existence of a mass gap between the known particles and the scale of the new dynamics, and whether these particles are weakly or strongly coupled to the new sector. The power counting is needed to define a hierarchy among the new-physics corrections. It then allows for a systematic approximation scheme based on a consistent expansion and truncation.

Quite generally, any relativistic quantum EFT is governed by expansions in both, inverse powers of a large mass scale $\Lambda$, and the number of loops. In the case of SMEFT, the corresponding expansion parameters can be taken as

$$\frac{E^2}{\Lambda^2} \qquad \text{and} \qquad \frac{1}{16\pi^2}, \tag{29}$$

with $E$ the energy scale of a given process and $1/16\pi^2$ the loop factor in four dimensions. Typically, $E$ will be a few times the electroweak scale $v = 246\,\text{GeV}$.

Usually in writing the SMEFT Lagrangian, only the expansion in $1/\Lambda$ is made explicit. Including terms up to order $1/\Lambda^2$, and assuming baryon- and lepton-number conservation, the Lagrangian can be written as

$$\mathcal{L}_{\text{SMEFT}} = \mathcal{L}_{\text{SM}} + \sum_i \frac{C_i}{\Lambda^2} Q_i, \tag{30}$$

where $\mathcal{L}_{\text{SM}}$ is the SM Lagrangian (terms of dimension 4), $Q_i$ are operators of dimension 6 [1,2] and $C_i$ are dimensionless coefficients. This EFT is defined with the particle content and the gauge symmetry of the SM, the baryon- and lepton-number conservation assumed here, and possibly further simplifying (symmetry) assumptions about the flavour structure in the dimension-6 terms with fermions.

We will now consider the power counting of SMEFT in more detail.

### 3.1 Example: Higgs production in gluon fusion

We start by considering the case of Higgs-boson production through gluon fusion, one of the most important processes in Higgs physics at the LHC. The lowest-order amplitude in SM

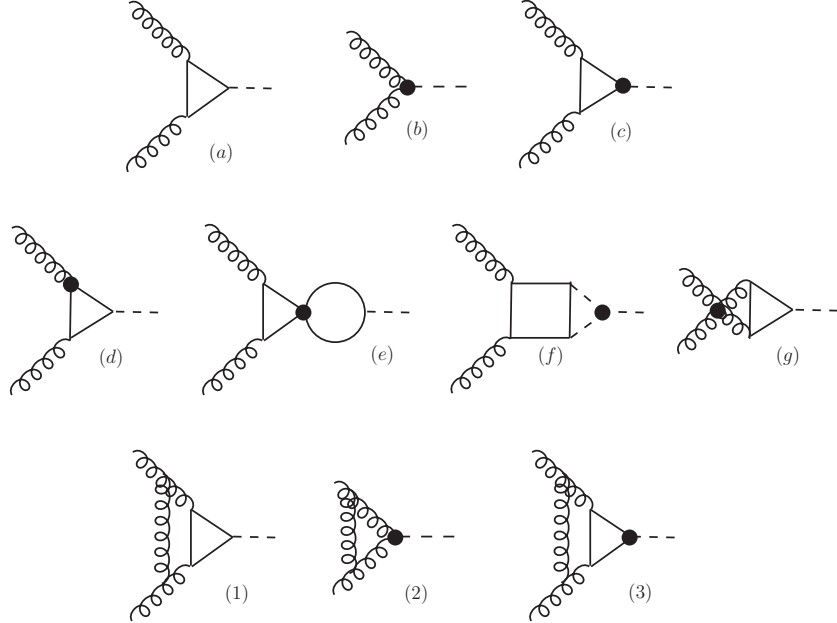

Figure 2: Higgs production through gluon fusion. (a): SM amplitude to lowest order (two diagrams with opposite fermion flow in the loop are understood). (b) – (g): Sample diagrams with insertions of dimension-6 operators (black dots) in SMEFT. (1) – (3): Examples of radiative corrections to the diagrams shown before.

perturbation theory is shown in Fig. 2 (a). We are interested in the corrections to this process of order $1/\Lambda^2$ in SMEFT. As illustrated in Fig. 2 (b) – (g), dimension-6 operators of almost all the classes defined in the Warsaw basis [2] do contribute to the process $gg \to h$. In particular (in the notation of [2])

$$
\begin{aligned}
(b): \quad & Q_{\varphi G} = \phi^\dagger \phi \, G^A_{\mu\nu} G^{A\mu\nu} \\
(c): \quad & Q_{u\varphi} = \phi^\dagger \phi \, \bar{q} u \tilde{\phi} \\
(d): \quad & Q_{uG} = \bar{q} \sigma^{\mu\nu} T^A u \, \tilde{\phi} G^A_{\mu\nu} \\
(e): \quad & Q_{uu} = \bar{u} \gamma_\mu u \, \bar{u} \gamma^\mu u, \dots \qquad \text{(4 - fermion operators with top)} \\
(f): \quad & Q_\varphi = (\phi^\dagger \phi)^3, \quad Q_{\varphi\Box} = \phi^\dagger \phi \Box \phi^\dagger \phi \\
(g): \quad & Q_G = f^{ABC} G^{A\nu}_\mu G^{B\rho}_\nu G^{C\mu}_\rho
\end{aligned}
\tag{31}
$$

A central question is: Which of these effects need to be included in a systematic treatment of the $1/\Lambda^2$ corrections? We see that the contributions in Fig. 2 (b) – (g) arise from diagrams with zero, one and two loops. This already suggests that loop counting should play a role in organizing the corrections in SMEFT. The counting of loops becomes unavoidable when we consider that, in practice, any amplitude, whether in the pure SM or in SMEFT, is computed in perturbation theory, which is an expansion in loop orders (equivalent to an expansion in powers of weak couplings). In fact, as illustrated in Fig. 2 (1) – (3), the perturbative expansion and the EFT expansion have to be combined in a certain way, consistent with the adopted power-counting rules.

The key to answering the above question therefore has to be based on a consideration of loop counting in SMEFT. Before we enter a discussion of this topic, we add a few remarks on trying to apply SMEFT without a systematic counting of loops.

- Suppose that, in order to be completely general, we want to take into account all possible effects from dimension-6 operators, as shown in Fig. 2 (b) – (g). To justify this despite the different loop orders, we think of the coefficients $C_i$ as arbitrary (dimensionless) numbers. However, such an approach would not be consistent: If arbitrary coefficients are allowed, there is no reason why e.g. some dimension-8 operator with a very large coefficient could not yield equally important effects as the dimension-6 corrections in Fig. 2. In this case the EFT treatment would break down. Specific power-counting assumptions about the $C_i$ are thus unavoidable.

- The most obvious choice of power counting would seem to be the one solely based on the canonical dimensions of the operators $Q_i$ in (30). The terms of dimension 6 are suppressed by two powers of the new-physics scale $\Lambda$, with coefficients taken to be $C_i = \mathcal{O}(1)$. Based on the explicit loop factors, diagram 2 (e) could therefore be neglected as subleading with respect to 2 (d). However, as we have demonstrated in Sec. 2, such a truncation would fail to correctly account for generic new-physics effects in the top sector. Specifically, the impact of any heavy resonance (weakly or strongly) coupled to top quarks would not be properly described by an EFT treatment that is supposed to be model-independent. Similarly, a strict application of $C_i = \mathcal{O}(1)$ would tell us that Fig. 2 (b) alone gives the leading correction for $gg \to h$ in the SMEFT because it is the only tree-level contribution at dimension 6. This is again in contradiction with typical scenarios of new physics [9, 10]. We conclude that the simple assignment $C_i = \mathcal{O}(1)$ for all coefficients in (30) is likewise not adequate as a consistent scheme to organize SMEFT. The missing ingredient is loop counting. The latter is part of a general power-counting prescription, to which we turn next.

## 3.2 Power counting

In this section we review the general power-counting rules for a relativistic EFT, which we can use for a systematic formulation of SMEFT. As stated at the beginning of Sec. 3 and illustrated with the example in 3.1, we need to count both powers of $E^2/\Lambda^2$ and loop factors $1/16\pi^2$ in general.

A convenient way to do this is to employ the canonical dimensions as well as the chiral dimensions of the terms in the EFT Lagrangian. This has been discussed in [11] and shown to be equivalent to well-known results on power counting in the literature [8, 12–14].

We consider a general relativistic EFT of scalars $\varphi$, gauge fields $A$ and fermions $\psi$ and assume that the theory has a cut-off scale $\Lambda$ (the scale of new physics). The EFT is valid at energies sufficiently below $\Lambda$. Let us *define* an energy scale $f$,

$$f \equiv \frac{\Lambda}{4\pi}. \tag{32}$$

We can view this as a reference scale, at which the EFT is a valid description of the relevant physics ($f \ll \Lambda$). For energies $E = f$ the parameter of the energy expansion, $E^2/\Lambda^2$, becomes $f^2/\Lambda^2 = 1/16\pi^2$, identical to a loop factor. We emphasize that the low-energy expansion is governed by $E^2/\Lambda^2$, with $E$ the actual energy of the process. This parameter is of course independent of the loop factor and $E$ may be several times larger or smaller than $f$. Introducing a reference energy $f$ simply amounts to a bookkeeping device, which treats the expansions in $E^2/\Lambda^2$ and $1/16\pi^2$ nominally on the same footing. The convenience of this will become apparent below.

Consider a general term in the EFT Lagrangian, schematically

$$\partial^{N_p} \varphi^{N_\varphi} A^{N_A} \psi^{N_\psi} \kappa^{N_\kappa} \,. \tag{33}$$

It is composed of a number of fields $\varphi$, $A$ and $\psi$, derivatives $\partial$, and some factors of weak couplings, generically denoted by $\kappa$.

The task of power counting is to estimate the size of the coefficient in front of (33). For this purpose both the canonical and the chiral dimension of (33), $d_c$ and $d_\chi$, are needed: The Lagrangian has canonical dimension 4, so the coefficient contains a factor $f^{4-d_c}$. Factors of $1/16\pi^2$ are counted by the loop order $L = (d_\chi - 2)/2$ (from the definition of $d_\chi$). The coefficient is therefore given by

$$C(d_c, d_\chi) = \frac{f^{4-d_c}}{(4\pi)^{d_\chi - 2}} \,. \tag{34}$$

By inspection of (33), the values of $d_c$ and $d_\chi$ are (see footnote 2)

$$d_c = N_p + N_\varphi + N_A + \frac{3}{2} N_\psi \,, \tag{35}$$

$$d_\chi = N_p + \frac{1}{2} N_\psi + N_\kappa \,. \tag{36}$$

We remark that the result (34) for the coefficient holds in general and independently of whether the EFT has a weakly or strongly coupled UV completion. However, the interpretation of $f$ is different in the two cases. For a weakly coupled UV completion $f$ is just the reference scale $f \equiv \Lambda/4\pi$, derived from $\Lambda$. In the case of strongly coupled UV physics, on the other hand, $f$ corresponds to a dynamical scale with its own physical meaning. For example, in chiral perturbation theory for pions in QCD, $f = f_\pi$ is the pion decay constant. Here $f$ is related to the QCD resonance scale $\Lambda$ through the NDA relation $\Lambda = 4\pi f$ [13], in correspondence with (32).

In the following, we will mostly focus on the application of (34) to SMEFT. It is then convenient to rewrite the coefficient (34) as

$$C(d_c, d_\chi) = \frac{1}{\Lambda^{d_c - 4}} \left( \frac{1}{16\pi^2} \right)^{(d_\chi - d_c)/2 + 1} \,. \tag{37}$$

Obviously, both $d_c$ and $d_\chi$ are required to obtain the power-counting estimate of a general operator coefficient. Powers of $1/\Lambda$ are simply dictated by the canonical dimension of the operator. Additionally, explicit loop factors are instead counted by

$$\frac{2 + d_\chi - d_c}{2} = \frac{2 + N_\kappa - N_F}{2} \,, \tag{38}$$

where we used (35) and (36), and introduced $N_F \equiv N_\varphi + N_A + N_\psi$, the total number of field factors in the operator. Thus, the loop factors are given by the difference of chiral and canonical dimension or, equivalently, the difference between the numbers of weak couplings and fields.

Some examples may serve to illustrate how (37) works. For instance, all terms in the SM Lagrangian (including $\mu^2 \phi^\dagger \phi$) have $d_c = 4$ and $d_\chi = 2$. Their coefficients are $C(4, 2) = 1$ by power counting, as has to be the case. We note that this assigns, in particular, a consistent power-counting size for the Higgs-mass operator in the SM Lagrangian. As mentioned in the Introduction, considering that $\phi^\dagger \phi$ has $d_c = 2$ is insufficient in this regard. We have to associate $\phi^\dagger \phi$ with a weak coupling carrying $d_\chi = 2$ to specify it as a leading-order term. Then its coefficient becomes $C(2, 2) = \Lambda^2/16\pi^2 = f^2 \ll \Lambda^2$, rather than the cutoff $\Lambda^2$, which would be inconsistent. As another example, a term like $g^2 \bar{\psi} \slashed{D} \psi$ has a coefficient $C(4, 4) = 1/16\pi^2$,

in agreement with its appearance as a self-energy counterterm for fermion $\psi$ at one loop. Further examples can easily be constructed.

When we deal with SMEFT corrections of dimension 6, (37) specializes to

$$C(6, d_\chi) = \frac{1}{\Lambda^2} \left( \frac{1}{16\pi^2} \right)^{(d_\chi - 4)/2} . \tag{39}$$

In order to determine the coefficient in (34), we need information about the number of weak couplings $N_\kappa$ associated with the operator in (33). Otherwise the power counting is incomplete. This is also evident from the fact that a SMEFT-like expansion and chiral perturbation theory have a different power counting, although formula (34) is valid for both.

To find $N_\kappa$ in (33), and therefore $d_\chi$ in (34), the power-counting prescription has to include a generic statement as to whether a field is weakly or strongly coupled to the heavy sector.[4] Different scenarios may be considered, but the corresponding assumptions should be formulated explicitly. In the following section, we will address this point and outline several possibilities.

## 3.3 Loop counting in SMEFT

We consider a generic extension of the SM by new physics at a scale $\Lambda \gg v$, which is weakly coupled to the SM fields and approximated by a Lagrangian that is renormalizable (in the traditional sense) at scale $\Lambda$. Weak coupling implies that the typical mass scale of heavy particles coincides with $\Lambda$. We take this as the standard scenario for SMEFT, which emerges at electroweak scales upon integrating out the large scale $\Lambda$.

It is conceivable that the assumption of weak coupling to the new physics is not fulfilled. However, this will have to be reflected in a modified power counting. For instance, if the Higgs sector is strongly coupled to the new physics, such as in composite-Higgs scenarios with a characteristic scale $f$, the electroweak-scale EFT will, in general, take the form of an electroweak chiral Lagrangian [8, 16–28]. When this EFT is explicitly expanded in $v/f$, a SILH-type [29] version of SMEFT is obtained [30]. In any case, the power counting of the EFT will be affected.

We now focus on the case of SMEFT under the assumption of weak coupling to the new-physics sector. This will, in part, amount to a review of results already obtained in [31] from an analysis of the weak-coupling case in SMEFT. We present it here using the convenient notion of chiral dimensions. A general discussion including operators of dimension 8 has been given in [32].

We assume that the UV completion of the SM at a scale $\Lambda$ is given by a renormalizable theory of bosons (scalars or gauge-fields) and fermions. The fields include the fermions $f$ and the bosons $b$ of the SM, as well as new, heavy fermions $F$ and bosons $B$, with mass of order $\Lambda$. The key assumption now is that $f$ and $b$ are weakly coupled to $F$ and $B$, that is with coupling strength of order unity. It is immaterial whether there exist nonrenormalizable interactions of $F$ and $B$ suppressed by scales parametrically still larger than $\Lambda$.

Denoting a generic fermion (boson) by $\Psi = f, F$ ($\beta = b, B$), only a limited number of renormalizable interactions exist at scale $\Lambda$, schematically $\bar{\Psi}\Psi\beta$, $\beta^3$, $\beta^4$, and $\beta^2\partial\beta$. Listing

---

[4]The exactly solvable model discussed in [15] gives an example of a heavy sector that may be either weakly or strongly coupled. Both cases are governed by the power-counting rules described here and illustrate their application.

the terms that couple light fields $f$, $b$ to heavy fields $F$, $B$ we have

$$
\begin{array}{lllllll}
\bar{\Psi}\Psi\beta: & \bar{f}fB & \bar{f}Fb & \bar{F}fb & \bar{f}FB & \bar{F}fB & \bar{F}Fb & [\kappa] \\
\beta^3: & b^2B & bB^2 & & & & & [\kappa\mu] \\
\beta^4: & b^3B & b^2B^2 & bB^3 & & & & [\kappa^2] \\
\beta^2\partial\beta: & b\partial bB & B\partial Bb & & & & & [\kappa]
\end{array}
\tag{40}
$$

In square brackets, we show for the terms in each line the factor of generic weak couplings $\kappa$. For the triple-boson terms $\mu$ indicates the mass scale required by dimensional analysis. This parameter may be a heavy or a light scale. From (40) we read off the number of weak couplings associated to fields and currents as building blocks of composite operators. We use the notation $A \sim \kappa^n$ if a building block $A$ comes with (at least) $n$ powers of weak couplings. We see that

$$
b \sim \kappa, \qquad b^2 \sim \kappa, \qquad b\partial b \sim \kappa, \qquad b^3 \sim \kappa^2.
\tag{41}
$$

For fermion bilinears, only scalar or vector currents can appear in the renormalizable vertices in (40) and hence may come with a single weak coupling. Therefore

$$
\bar{f}\Gamma f \sim \kappa \quad \text{for} \quad \Gamma = 1, \gamma^\mu, \qquad \bar{f}\sigma^{\mu\nu}f \sim \kappa^2.
\tag{42}
$$

We next write down the different classes of dimension-6 operators in SMEFT, following the notation of [2]. Supplementing the operators with the minimum number of weak coupling factors, according to the considerations above, we find

$$
\kappa^4(\phi^\dagger\phi)^3, \qquad \kappa^2(\phi^\dagger D\phi)^2, \qquad \kappa^3\phi^\dagger\phi\,\bar{\psi}\phi\psi, \qquad \kappa^2\phi^\dagger D\phi\,\bar{\psi}\psi, \qquad \kappa^2(\bar{\psi}\psi)^2.
\tag{43}
$$

For the first term, a minimum of four weak couplings is required, e.g. two factors of $\kappa^2$, one for each $b^3$ part, or else three $\kappa$s from three $b^2$ terms coupled to heavy scalars, times another $\kappa$ from the coupling of those three heavy fields. The assignment of weak couplings to the remaining four operators is obvious. Finally, the operator classes with SM field strength factors read

$$
\kappa^3 X_\mu{}^\nu X_\nu{}^\lambda X_\lambda{}^\mu, \qquad \kappa^4\phi^\dagger\phi\,X_{\mu\nu}X^{\mu\nu}, \qquad \kappa^4\bar{\psi}\sigma_{\mu\nu}X^{\mu\nu}\phi\psi.
\tag{44}
$$

Here the first operator comes with three gauge couplings, connecting the gauge fields to the heavy sector that has been integrated out. Similarly, the second operator has two gauge couplings associated to the two field strengths, two more weak couplings are needed to connect the $\phi^\dagger\phi$ part. In the third operator, $\kappa^2$ comes with the fermionic tensor current, and one $\kappa$ each from $X^{\mu\nu}$ and $\phi$.

The chiral dimension of the SMEFT operators can now be read off immediately: $d_\chi = 4$ for all terms in (43) and $d_\chi = 6$ for the terms in (44). Using the power-counting formula (39), this implies coefficients of order $1/\Lambda^2$ for the operators in (43) and of order $1/16\pi^2\Lambda^2$ for those in (44). In short, assuming weak coupling to the heavy sector, the SMEFT operators of the Warsaw basis with gauge field strength factors carry an extra loop suppression [2, 31].

This loop-counting hierarchy among the operator coefficients fits naturally with the organization of SMEFT corrections in terms of both canonical and chiral dimensions. As an example, let us again consider Higgs production in gluon fusion discussed in Sec. 3.1. Combining the power-counting size of the operator coefficients, $1/\Lambda^2$ for the coefficients of (43) and $1/16\pi^2\Lambda^2$ for the coefficients of (44), with the explicit loop factor from the diagram topology, we find a clear ordering of the contributions to $gg \to h$ shown in Fig. 2 $(a)$ – $(g)$:

$$
\frac{1}{16\pi^2}:(a), \qquad \frac{1}{16\pi^2\Lambda^2}:(b),(c), \qquad \frac{1}{(16\pi^2)^2\Lambda^2}:(d),(e),(f), \qquad \frac{1}{(16\pi^2)^3\Lambda^2}:(g).
\tag{45}
$$

The dominant contribution is the SM amplitude, Fig. 2 (*a*). It comes with a loop factor but is unsuppressed in $1/\Lambda$. The remaining terms in (45) are SMEFT corrections from the dimension-6 Lagrangian. They all carry a factor of $1/\Lambda^2$ but enter, effectively, at different loop orders. Note that (*b*) and (*c*) enter at the same order, even though the former is a tree and the latter a loop diagram. Similarly, (*d*) – (*f*) have the same power-counting size, despite their different topology. On the other hand, (*e*) and (*g*) have the same (two-loop) topology, but count at different orders. We also remark that the magnetic-moment type vertex correction in Fig. 2 (*d*) and the 4-fermion contribution in (*e*) entering at the same order is in accordance with the detailed discussion in Sec. 2 and Sec. 4 below.

The systematic power counting by canonical *and* chiral dimensions, illustrated in (45), provides us with a consistent truncation scheme of the SMEFT expansion. For instance, if we aim for the leading new-physics effects, only the corrections of the same order as (*b*) and (*c*) need to be retained. Those are of order $1/\Lambda^2$ relative to the SM term in (*a*). All others, (*d*) – (*g*), are higher order in the loop counting and can be dropped. We emphasize that all issues concerning the treatment of the various SMEFT contributions to $gg \to h$ mentioned in Sec. 3.1 are resolved in the present scheme. A more detailed description of the leading effects will be given in Sec. 3.4 below.

Finally, radiative corrections, e.g. those shown in Fig. 2 (1) – (3), may be incorporated to any desired accuracy if needed, consistently with the counting scheme discussed above.

There are many more processes and observables in high-energy collider physics to which the above power counting can be applied. It will be of use, in particular, to put the focus on the potentially leading SMEFT effects. In practice, the consistent truncation of corrections will also help to reduce the number of free parameters in a SMEFT analysis in a meaningful way. Examples for possible applications can be found in the literature. Recent studies of single-Higgs production in SMEFT include [33–37]. Higgs-boson pair production at NLO and beyond has been investigated in [38–46]. In [43] the systematic loop counting for SMEFT has already been discussed for this particular application. A study of top-quark pair production via gluon fusion in SMEFT, following the power counting discussed here, can be found in [47]. The pattern of SMEFT effects in $gg \to h$ or $h \to gg$ is similar in $h \to \gamma\gamma$ decay. The latter process has been treated for instance in [48–50]. Detailed calculations of $Zh$ production in $pp$ collisions including SMEFT corrections have recently appeared in [51, 52].

### 3.4 Amplitude for $gg \to h$ with leading dim-6 corrections in SMEFT

In the previous section, we have identified the dimension-6 contributions in Fig. 2 (*b*) and (*c*) as the leading SMEFT corrections to Higgs-boson production in gluon fusion. To be more specific, we spell out in the following the $gg \to h$ amplitude with these corrections included. In the Warsaw basis [2], the terms relevant to the process $gg \to h$ read

$$
\begin{aligned}
\Delta\mathcal{L}_{\text{Warsaw}} = {} & \frac{C_{H\square}}{\Lambda^2}(\phi^\dagger \phi)\square(\phi^\dagger\phi) + \frac{C_{HD}}{\Lambda^2}(\phi^\dagger D_\mu \phi)^*(\phi^\dagger D^\mu \phi) \\
& + \left(\frac{C_{uH}}{\Lambda^2}\phi^\dagger \phi\, \bar{q}_L \tilde{\phi}\, t_R + h.c.\right) + \frac{C_{HG}}{\Lambda^2}\phi^\dagger \phi\, G^A_{\mu\nu}G^{A,\mu\nu}.
\end{aligned}
\tag{46}
$$

Expanding the Higgs doublet in eq. (46) around its vacuum expectation value and applying a field redefinition for the physical Higgs boson

$$
h \to h + \frac{v^2}{\Lambda^2}C_{H,kin}\left(h + \frac{h^2}{v} + \frac{h^3}{3v^2}\right), \qquad \text{where} \quad C_{H,kin} \equiv C_{H\square} - \frac{1}{4}C_{HD},
\tag{47}
$$

the Higgs kinetic term acquires its canonical form (up to $\mathcal{O}\left(\Lambda^{-4}\right)$ terms).

The Lagrangian for anomalous Higgs couplings relevant to Higgs boson production in gluon fusion can in general be parametrised by [43]

$$\Delta\mathcal{L}_h = -m_t\, c_t \frac{h}{v}\, \bar{t}\, t + \frac{\alpha_s}{8\pi} c_{ggh} \frac{h}{v}\; G^A_{\mu\nu} G^{A,\mu\nu}. \tag{48}$$

Comparing the coefficients of the corresponding terms in the Lagrangians (46) and (48) leads to the following relations between the Higgs couplings $c_t$ and $c_{ggh}$ and the SMEFT coefficients in the Warsaw basis:

$$c_t = 1 + \frac{v^2}{\Lambda^2} C_{H,kin} - \frac{v^2}{\Lambda^2} \frac{v}{\sqrt{2}m_t} C_{uH} \equiv 1 + \delta_{c_t}, \qquad c_{ggh} = \frac{v^2}{\Lambda^2} \frac{8\pi}{\alpha_s} C_{HG}. \tag{49}$$

In the Warsaw basis, three dimension-6 operators contribute to the coefficient $\delta_{c_t}$. Note that both $c_t$ and $c_{ggh}$ are invariant under QCD renormalization. This is not the case for the SMEFT coefficients $C_{uH}$ and $C_{HG}$ [53–55]. We also remark that $C_{H,kin}$, $C_{uH}$ and $C_{HG}$ have chiral dimension $d_\chi = 2$, 3 and 4, respectively (Sec 3.3). In their usual definition, they are thus not homogeneous in $d_\chi$. On the other hand, $d_\chi = 2$ for both $\delta_{c_t}$ and $c_{ggh}$, assuming weakly-coupled SMEFT power counting.

For the consistency of the present treatment, amplitudes should be expanded through order $v^2/\Lambda^2$ and higher orders omitted. Indeed, for a typical value of $\Lambda = 3$ TeV, $v^2/\Lambda^2 \approx 7 \cdot 10^{-3}$. This is a small parameter, even neglecting possible further suppression from coupling factors. Fit results for $\delta_{c_t}$ and $c_{ggh}$ that are compatible with zero, but still allow for deviations at the 10 – 20% level, likely suggest that data are not yet sensitive to new-physics effects, rather than requiring the inclusion of dimension-8 operators.

The amplitude for the process $g(p_1,\mu) + g(p_2,\nu) \to h(p_3)$ can be decomposed as

$$\begin{aligned} \mathcal{M}_{AB} &= \delta_{AB}\, \epsilon_\mu(p_1)\epsilon_\nu(p_2)\, \mathcal{M}^{\mu\nu}, \\ \mathcal{M}^{\mu\nu} &= \frac{\alpha_s}{8\pi v} \mathcal{F}_1\, T^{\mu\nu}. \end{aligned} \tag{50}$$

Here $A$, $B$ are colour indices, $\epsilon_\mu, \epsilon_\nu$ are the gluon polarization vectors, and

$$T^{\mu\nu} = g^{\mu\nu} - \frac{p_1^\nu p_2^\mu}{p_1 \cdot p_2}. \tag{51}$$

The form factor $\mathcal{F}_1$ is given by the expression [56–59]

$$\mathcal{F}_1 = 2s_{12}\Big\{ (1+\delta_{c_t})\tau_t\, [1 + (1-\tau_t)f(\tau_t)] + c_{ggh} \Big\}, \tag{52}$$

where $\tau_t = 4m_t^2/s_{12}$, $s_{12} = 2p_1 \cdot p_2 = m_h^2$ for on-shell Higgs boson production and

$$f(\tau_t) = -\frac{1}{2}s_{12}C_{12} = \begin{cases} \arcsin^2 \dfrac{1}{\sqrt{\tau_t}}, & \text{for } \tau_t \geq 1, \\[2mm] -\dfrac{1}{4}\left[ \log\dfrac{1+\sqrt{1-\tau_t}}{1-\sqrt{1-\tau_t}} - i\pi \right]^2, & \text{for } \tau_t < 1, \end{cases}$$

$$C_{12} = \int \frac{d^4k}{i\pi^2} \frac{1}{(k^2 - m_t^2)\big[(k+p_1)^2 - m_t^2\big]\big[(k+p_1+p_2)^2 - m_t^2\big]}. \tag{53}$$

At relative order $v^2/\Lambda^2$, in addition to the effects considered so far, the amplitude for $gg \to h$ also receives corrections from the operators $Q^{(3)}_{\varphi l}$ and $Q^{1221}_{ll}$ [2]. These operators modify the muon decay rate, from which the Fermi constant $G_F$ is extracted in order to determine $v$ in

(50). Defining $G_{F0} = 1/(\sqrt{2}v^2)$ and denoting by $G_F$ the Fermi constant measured from muon decay, we may write [50, 60]

$$G_{F0} = G_F (1 - 2\delta_G),\qquad(54)$$

with

$$2\delta_G = \frac{v^2}{\Lambda^2}\left(C_{\varphi l,1}^{(3)} + C_{\varphi l,2}^{(3)} - C_{1221}^{ll}\right)\qquad(55)$$

in the notation of [2]. Here the numerical subscripts are generation indices. This correction to (50) from new physics in the first two lepton generations is numerically small. Electroweak fits constrain $2\delta_G$ to be safely below the percent level [61]. Neglecting it leaves $\delta_{c_t}$ and $c_{ggh}$ as the relevant corrections to $gg \to h$ at leading order in SMEFT.

## 4 Example for SMEFT in a UV model: $u\bar{u} \to t\bar{t}$ via gluon exchange in the 2HDM

In the following section we generalize the EFT features illustrated with the toy model of Sec. 2 to a more complete scenario in the context of the full SM. For this purpose, we employ a Two-Higgs-Doublet model (2HDM) [62–64] as the UV completion of SMEFT. In contrast to the SM, this model features not one, but two independent scalar SU(2) doublets $\Phi_1$ and $\Phi_2$. Interpreting the 2HDM as an adequate UV extension of the SM, it should be possible to match the former to the latter at the electroweak scale in terms of higher dimensional SMEFT operators.

The potential of the 2HDM is given by the most general expression allowed by symmetries

$$\begin{aligned}
V(\Phi_1,\Phi_2) &= m_{11}^2\Phi_1^\dagger\Phi_1 + m_{22}^2\Phi_2^\dagger\Phi_2 - \left[m_{12}^2\Phi_1^\dagger\Phi_2 + \text{h.c.}\right] + \frac{1}{2}\lambda_1\left(\Phi_1^\dagger\Phi_1\right)^2 + \frac{1}{2}\lambda_2\left(\Phi_2^\dagger\Phi_2\right)^2 \\
&\quad + \lambda_3\left(\Phi_1^\dagger\Phi_1\right)\left(\Phi_2^\dagger\Phi_2\right) + \lambda_4\left(\Phi_1^\dagger\Phi_2\right)\left(\Phi_2^\dagger\Phi_1\right) \\
&\quad + \left\{\frac{1}{2}\lambda_5\left(\Phi_1^\dagger\Phi_2\right)^2 + \left[\lambda_6\left(\Phi_1^\dagger\Phi_1\right) + \lambda_7\left(\Phi_2^\dagger\Phi_2\right)\right]\left(\Phi_1^\dagger\Phi_2\right) + \text{h.c.}\right\}.
\end{aligned}\qquad(56)$$

Imposing CP-invariance, we take the coefficients $m_{12}^2, \lambda_5, \lambda_6$ and $\lambda_7$ to be real. Without loss of generality, we allow both doublets to pick up a vacuum expectation value $v_i > 0$ $(i = 1, 2)$ in the lower component

$$\Phi_i = \begin{pmatrix} \phi_i^+ \\ \frac{1}{\sqrt{2}}\left[v_i + \rho_i + i\eta_i\right] \end{pmatrix},\qquad(57)$$

where $\phi_i^+$ is a complex scalar ($\phi_i^-$ being its hermitian conjugate) and $\rho_i$ and $\eta_i$ are real scalars. We identify the physical states by diagonalizing the mass matrices and find that out of in total eight scalar degrees of freedom, three are Goldstone modes ($G^\pm$ and $G$) which subsequently become the longitudinal degrees of freedom of the $W^\pm$ and $Z$ bosons. In addition, there are two massive neutral scalars $h$ and $H$, a massive pseudoscalar $A$ and a massive charged scalar $H^\pm$ with masses $m_h$, $m_H$, $m_A$ and $m_{H^\pm}$, respectively. In terms of these states, the doublets are given by

$$\Phi_1 = \begin{pmatrix} c_\beta G^+ - s_\beta H^+ \\ \frac{1}{\sqrt{2}}\left[v_1 + c_\alpha H - s_\alpha h + ic_\beta G - is_\beta A\right] \end{pmatrix},\qquad(58)$$

$$\Phi_2 = \begin{pmatrix} s_\beta G^+ + c_\beta H^+ \\ \frac{1}{\sqrt{2}}\left[v_2 + s_\alpha H + c_\alpha h + is_\beta G + ic_\beta A\right] \end{pmatrix},\qquad(59)$$

where the shorthand notations $c_\varphi \equiv \cos\varphi$ and $s_\varphi \equiv \sin\varphi$ have been introduced. The mixing angle $\beta$ is defined via $\tan\beta = v_2/v_1$. Explicit expressions for the non-vanishing masses as well as for $v_1$, $v_2$ and the mixing angle $\alpha$ in terms of the parameters in (56) can be found in [63]. For the sake of our following arguments, we choose a general *type-II Yukawa sector* given by the Lagrangian [62]

$$\mathcal{L}_Y = -\bar{q}_L \Phi_1 Y_d d_R - \bar{q}_L \tilde{\Phi}_2 Y_u u_R + \text{h.c.}, \tag{60}$$

where $\tilde{\Phi}_i \equiv i\sigma_2 \Phi_i^*$. The Yukawa-coupling matrices $Y_d$ and $Y_u$ act in flavour space and $\bar{q}_L$, $d_R$ and $u_R$ are the usual left- and right-handed quark fields with flavour indices suppressed. Throughout the discussion, we neglect effects of the CKM-matrix. Choosing the couplings between the scalar sector and the fermions in this manner has the advantage of being particularly transparent in the so-called *decoupling limit* (see (63) below).

Rotating the scalar doublets by the angle $\beta$, we can shift the vacuum expectation value to only one doublet. This rotated basis is known as the *Higgs basis* in the literature [65]. The rotated doublets $H_1$ and $H_2$ have the explicit form

$$H_1 = \begin{pmatrix} G^+ \\ \frac{1}{\sqrt{2}} \left[ v + c_{\beta-\alpha} H + s_{\beta-\alpha} h + iG \right] \end{pmatrix}, \tag{61}$$

$$H_2 = \begin{pmatrix} H^+ \\ \frac{1}{\sqrt{2}} \left[ -s_{\beta-\alpha} H + c_{\beta-\alpha} h + iA \right] \end{pmatrix}, \tag{62}$$

where $v = \sqrt{v_1^2 + v_2^2} = 246\,\text{GeV}$ can be identified as the electroweak scale.

It is possible to choose the parameters in such a way that the physical masses follow the hierarchy pattern [63]

$$m_h^2 \ll m_H^2, m_A^2, m_{H^\pm}^2 \simeq m_S^2, \tag{63}$$

where $m_S \gg v$. An explicit expression for $m_S$ in terms of $\beta$ and the parameters in the potential is given in [63]. The decoupling limit mentioned above implies taking $s_{\beta-\alpha} \to 1$ and $c_{\beta-\alpha} \to 0$. In this limit, the couplings of $h$ become identical to those of the SM Higgs, which suggests interpreting $H_1$ as the SM Higgs doublet. The remaining doublet $H_2$ then only contains heavy fields and can be integrated out to obtain a low-energy theory at the electroweak scale $v$. Its effects are then entirely encoded in the Wilson coefficients of higher dimensional local operators (SMEFT).

## 4.1 Top-down EFT for $u\bar{u} \to t\bar{t}$

Similar to the toy model of Sec. 2, we analyze the process $u(k_1)\bar{u}(k_2) \to t(p_1)\bar{t}(p_2)$ via s-channel gluon exchange. The relevant diagrams are the same as in the toy model in Fig. 1 with the internal photon replaced by a gluon. We take only the top quark as massive, which implies that $Y_t = \sqrt{2} m_t \csc\beta/v$ is the only non-vanishing Yukawa coupling matrix element and that the heavy states couple exclusively to third-generation quarks. Defining $g = m_t \cot\beta/v$, the relevant interaction Lagrangian is given by

$$\mathcal{L}_{int} = g\bar{t}tH + g\bar{t}i\gamma_5 tA + \sqrt{2}g\bar{t}_R b_L H^+ + \text{h.c.}, \tag{64}$$

where $t$ and $b$ are the Dirac fields of the top- and bottom quarks with $t_{R/L} = P_{R/L}t$, etc. and $P_{R/L}$ the right- or left-handed projector, respectively. With the notation of Sec. 2, the correction to the amplitude can be written as

$$\delta\mathcal{A} = i\frac{g_s^2}{q^2} \bar{v}(k_2)\gamma_\mu T^A u(k_1) \bar{u}(p_1)\delta\Gamma^\mu T^A v(p_2), \tag{65}$$

where $T^A = \lambda^A/2$ are the generators of SU(3) with $\lambda^A$ the Gell-Mann matrices and $g_s$ is the QCD coupling constant. Here and in the following, we strictly work at order $g_s^2$ and subsequently drop terms of higher order without further comments.

In the full theory, the relevant diagrams are displayed in Fig. 1 (b) and (c), where the dashed line corresponds to either $H$, $A$ or $H^\pm$. Summing up all three contributions, we end up with

$$\delta\Gamma^\mu = \frac{g^2}{16\pi^2}\frac{1}{m_S^2}\left[ m_t i\sigma^{\mu\nu}q_\nu + \left(\frac{2}{9} - \frac{2}{3}\ln\frac{q^2}{m_S^2} + i\frac{2\pi}{3}\right)q^2\gamma^\mu P_R \right.$$
$$\left. - \left(\frac{2}{3}\ln\frac{m_t^2}{m_S^2} + \frac{8}{9} + 2h_1(z)\right)q^2\gamma^\mu \right], \tag{66}$$

where on-shell renormalization of the top quark has been employed as in [66] and we expanded to first order in $1/m_S^2$. Pure gauge terms proportional to $q^\mu$ have been dropped since they do not contribute to physical processes.

Expression (66) can be reproduced by an effective field theory specified by the Lagrangian

$$\mathcal{L}_{eff} = \mathcal{L}_{eff}^{tree} + \mathcal{L}_{eff}^{loop} = \sum_{i=1}^{4}\frac{C_i}{m_S^2}Q_i, \tag{67}$$

where

$$\mathcal{L}_{eff}^{tree} = \frac{C_1}{m_S^2}\bar{t}_R q_L \bar{q}_L t_R \tag{68}$$

arises, with $C_1 = 2g^2$, when the heavy fields are integrated out at tree level and

$$\mathcal{L}_{eff}^{loop} = \frac{C_2}{m_S^2}D^\mu G_{\mu\nu}^A \bar{t}_R T^A \gamma^\nu t_R + \frac{C_3}{m_S^2}D^\mu G_{\mu\nu}^A \bar{t}T^A\gamma^\nu t + \frac{C_4}{m_S^2}m_t G_{\mu\nu}^A \bar{t}T^A\sigma^{\mu\nu}t \tag{69}$$

is generated at one loop, where $G_{\mu\nu}^A$ is the gluonic field strength tensor and $q_L$ the left-handed third-generation quark doublet. The loop diagram associated with $\mathcal{L}_{eff}^{tree}$ is displayed in Fig. 1 (e) and gives

$$\delta\Gamma_{Q_1}^\mu = \frac{C_1}{16\pi^2 m_S^2}\left[\left(\frac{5}{9} - \frac{1}{3}\ln\frac{q^2}{\mu^2} + i\frac{\pi}{3}\right)q^2\gamma^\mu P_R - \left(\frac{1}{3}\ln\frac{m_t^2}{\mu^2} + h_1(z)\right)q^2\gamma^\mu\right], \tag{70}$$

and the tree-contributions from $\mathcal{L}_{eff}^{loop}$ in Fig. 1 (d) read

$$\delta\Gamma_{Q_{2-4}}^\mu = \frac{1}{g_s}\left[\frac{C_2}{m_S^2}q^2\gamma^\mu P_R + \frac{C_3}{m_S^2}q^2\gamma^\mu - \frac{2C_4}{m_S^2}m_t i\sigma^{\mu\nu}q_\nu\right]. \tag{71}$$

Performing the matching procedure reveals that the coefficients are given by

$$C_1 = 2g^2, \qquad C_2 = C_3 = -\frac{g_s g^2}{16\pi^2}\left(\frac{2}{3}\ln\frac{\mu^2}{m_S^2} + \frac{8}{9}\right), \qquad C_4 = -\frac{g_s g^2}{16\pi^2}\frac{1}{2}. \tag{72}$$

As in Sec. 2.2, the dependence on $\mu$ cancels when both contributions are added and the full result is restored.

Note that the four local operators in (67) – (69) can be matched to the Warsaw basis [2] by virtue of Fierz identities and the equations of motion for the gluons. Dropping terms that do not contribute to the process at hand, the relevant expressions are given by (here the upper indices display generation labels)

$$Q_1 \longrightarrow -\left(Q_{qu}^{(8)3333} + \frac{1}{6}Q_{qu}^{(1)3333}\right),\tag{73}$$

$$Q_2 \longrightarrow g_s\left(Q_{qu}^{(8)1133} + \frac{1}{4}Q_{uu}^{1331} + \frac{1}{4}Q_{uu}^{3113} - \frac{1}{12}Q_{uu}^{1133} - \frac{1}{12}Q_{uu}^{3311}\right),\tag{74}$$

$$Q_3 \longrightarrow g_s\left(Q_{qu}^{(8)3311} + Q_{qu}^{(8)1133} + \frac{1}{4}Q_{uu}^{1331} - \frac{1}{12}Q_{uu}^{1133} + \frac{1}{8}Q_{qq}^{(3)1331} + \frac{1}{8}Q_{qq}^{(1)1331}\right.$$
$$\left.-\frac{1}{12}Q_{qq}^{(1)1133} + \frac{1}{4}Q_{uu}^{3113} - \frac{1}{12}Q_{uu}^{3311} + \frac{1}{8}Q_{qq}^{(3)3113} + \frac{1}{8}Q_{qq}^{(1)3113} - \frac{1}{12}Q_{qq}^{(1)3311}\right),\tag{75}$$

$$Q_4 \longrightarrow \frac{\sqrt{2}m_t}{v}\left(Q_{uG}^{33} + Q_{uG}^{*33}\right).\tag{76}$$

Note that in the Warsaw basis, the operators $Q_2$ and $Q_3$ introduce an extra factor of $g_s$. This has to be so, as treating the four-fermion operators introduced in this manner on the same footing as $Q_1$ would spoil the underlying systematic expansion in $g_s$. This is analoguous to (16) in the toy model. It is now straightforward to identify the relevant Wilson coefficients of the Warsaw basis operators to order $g_s^2$ for the process at hand. The explicit expressions are given below.

### 4.2  Bottom-up SMEFT calculation

Without referring to the UV model, we would have started with a new-physics scale $\Lambda$ and the complete set of Warsaw basis operators that are relevant for the process under consideration. We have to distinguish between four-fermion contributions entering at tree or one-loop level, respectively. The tree contribution is given by the plain four-fermion vertex (here $\bar{v} = \bar{v}(k_2)$, $u = u(k_1)$, $\bar{u} = \bar{u}(p_1)$ and $v = v(p_2)$)

$$\begin{aligned}
\delta\mathcal{A}_{tree} = \frac{i}{\Lambda^2}\Big( &2\left(C_{qq}^{(1)1133} + C_{qq}^{(3)1133}\right)\bar{v}\gamma_\mu P_L u\bar{u}\gamma^\mu P_L v \\
&-2\left(C_{qq}^{(1)1331} + C_{qq}^{(3)1331}\right)\bar{v}\gamma_\mu P_L v\bar{u}\gamma^\mu P_L u \\
&+2C_{uu}^{(1)1133}\bar{v}\gamma_\mu P_R u\bar{u}\gamma^\mu P_R v - 2C_{uu}^{(1)1331}\bar{v}\gamma_\mu P_R v\bar{u}\gamma^\mu P_R u \\
&+C_{qu}^{(1)1133}\bar{v}\gamma_\mu P_L u\bar{u}\gamma^\mu P_R v + C_{qu}^{(1)3311}\bar{v}\gamma_\mu P_R u\bar{u}\gamma^\mu P_L v \\
&-C_{qu}^{(1)1331}\bar{v}\gamma_\mu P_L v\bar{u}\gamma^\mu P_R u - C_{qu}^{(1)3113}\bar{v}\gamma_\mu P_R v\bar{u}\gamma^\mu P_L u \\
&+C_{qu}^{(8)1133}\bar{v}T^A\gamma_\mu P_L u\bar{u}T^A\gamma^\mu P_R v + C_{qu}^{(8)3311}\bar{v}T^A\gamma_\mu P_R u\bar{u}T^A\gamma^\mu P_L v \\
&-C_{qu}^{(8)1331}\bar{v}T^A\gamma_\mu P_L v\bar{u}T^A\gamma^\mu P_R u - C_{qu}^{(8)3113}\bar{v}T^A\gamma_\mu P_R v\bar{u}T^A\gamma^\mu P_L u\Big),
\end{aligned}\tag{77}$$

whereas the one-loop contribution yields (Fig. 1 (e))

$$\delta\Gamma_{loop}^\mu = -\frac{1}{16\pi^2\Lambda^2}(q^2\gamma^\mu F_1(q^2) + m_t i\sigma^{\mu\nu}q_\nu F_2(q^2)),\tag{78}$$

with

$$\begin{aligned}
F_1 = &\left(\frac{5}{9} - \frac{1}{3}\ln\frac{q^2}{\mu^2} + i\frac{\pi}{3}\right)\left(\left(C_{ud}^{(8)3333} + C_{qu}^{(8)3333}\right)P_R + \left(C_{qd}^{(8)3333} + 8C_{qq}^{(3)3333}\right)P_L\right) \\
&-\left(\frac{1}{3}\ln\frac{m_t^2}{\mu^2} + h_1(z)\right)\left(C_{qu}^{(8)3333} + 4\left(C_{qq}^{(1)3333} + C_{qq}^{(3)3333}\right)P_L + 4C_{uu}^{3333}P_R\right) \\
&-\frac{4}{3}\left(\left(C_{qq}^{(1)3333} + 3C_{qq}^{(3)3333}\right)P_L + C_{uu}^{3333}P_R\right),
\end{aligned}$$

$$F_2 = 2\left(C_{qu}^{(1)3333} - \frac{1}{6}C_{qu}^{(8)3333}\right). \tag{79}$$

In addition, the chromomagnetic operator enters at tree-level as before (Fig. 1 (d)). Its contribution is given by

$$\delta\Gamma_{uG}^{\mu} = -\frac{\sqrt{2}v}{g_s\Lambda^2}i\sigma^{\mu\nu}q_{\nu}\left(C_{uG}^{*33}P_L + C_{uG}^{33}P_R\right). \tag{80}$$

Note that we have implicitly assumed the new-physics sector to couple to the third particle generation only as we neglected generation mixing four-fermion operators in the one-loop contribution. For a comparison to the previous section, it is advantageous to rewrite the four-fermion tree contribution by virtue of Fierz identities like $\bar{v}(k_2)\gamma_\mu P_L v(p_2)\bar{u}(p_1)\gamma^\mu P_L u(k_1) = -\bar{v}(k_2)\gamma_\mu P_L u(k_1)\bar{u}(p_1)\gamma^\mu P_L v(p_2)$ and $2T_{ab}^A T_{cd}^A = \delta_{ad}\delta_{bc} - \delta_{ab}\delta_{cd}/3$.

Comparing (77) – (80) with the top-down results in Sec. 4.1 reveals that when identifying $\Lambda$ with $m_S$, the non-vanishing SMEFT Wilson coefficients are given by

$$C_{qu}^{(8)3333} = 6C_{qu}^{(1)3333} = -2g^2, \tag{81}$$

$$2C_{uu}^{1331} = 2C_{uu}^{3113} = -6C_{uu}^{1133} = -6C_{uu}^{3311} = C_{qu}^{(8)3311} = \frac{1}{2}C_{qu}^{(8)1133} = 8C_{qq}^{(3)1331}$$

$$= 8C_{qq}^{(3)3113} = -12C_{qq}^{(1)1133} = -12C_{qq}^{(1)3311} = 8C_{qq}^{(1)1331} = 8C_{qq}^{(1)3113}$$

$$= -\frac{g_s^2 g^2}{16\pi^2}\left(\frac{2}{3}\ln\frac{\mu^2}{m_S^2} + \frac{8}{9}\right), \tag{82}$$

$$C_{uG}^{33} = C_{uG}^{*33} = -\frac{g_s g^2 m_t}{16\pi^2 v}\frac{1}{\sqrt{2}}. \tag{83}$$

The $\mu$-dependence matches the known results for the renormalization-group equations in SMEFT [53–55].

# 5 Conclusions

The effective field-theory approach to physics beyond the Standard Model requires a minimal set of assumptions about the relation between the low-energy and high-energy regimes, which is contained in the EFT power counting rules. While the Lagrangian of non-linear EFTs such as chiral perturbation theory is systematically ordered based on a loop expansion, the power counting in effective theories such as SMEFT is organized in terms of canonical dimensions.

We have discussed specific examples to show that a power counting scheme relying on canonical dimensions alone may lead to inconsistencies within the perturbative expansion. Starting with a toy model involving a heavy scalar singlet, we have illustrated our arguments by an explicit calculation, comparing both bottom-up and top-down EFT with the full theory. Furthermore, we deployed the power counting in combination with loop counting in SMEFT in detail. We applied it to Higgs production in gluon fusion as well as an example within a two-Higgs-doublet model. Our formal considerations as well as the examples clearly suggest that a consistent treatment should include the counting of loop orders, conveniently described by chiral dimensions, along with the counting of canonical dimensions in SMEFT. Variations of the counting scheme we propose can, of course, be constructed, but the approximations and counting rules should always be explicitly specified. For example, the SMEFT may be replaced by HEFT, which follows a different power counting. In addition, further assumptions, such as minimal flavour violation, can be used to organize the flavour sector, which however is a separate topic and beyond the scope of our paper.

# Acknowledgements

We thank Marius Höfer and Michael Trott for interesting discussions.

**Funding information** The work of G.B. was supported in part by the Deutsche Forschungsgemeinschaft (DFG, German Research Foundation) under grant BU 1391/2-2 (project number 261324988) and by the DFG under Germany's Excellence Strategy - EXC-2094 - 390783311. The work of G.H. was supported by the Deutsche Forschungsgemeinschaft (DFG, German Research Foundation) under grant 396021762 - TRR 257. Ch. M.-S. is supported by a Fellowship of the Studienstiftung des deutschen Volkes (German Academic Scholarship Foundation).

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
