# Peer review of "Loop counting matters in SMEFT"

_SciPost Physics, doi:SciPost Phys. 15, 088 (2023)_

## Round 1 · Referee Report · Anonymous (Referee 1) · 2022-8-25

Report

The authors discuss the role of loop suppressions in the matching between beyond-SM theories and dimension-6 effective operators in the Standard Model Effective Field Theory (SMEFT). They argue that these suppressions should be accounted for in the SMEFT power counting and they propose a methodology to do so by using the notion of chiral dimensions.

The methodology they suggest goes in the direction of making the well-known tree/loop operator classification, initially proposed by Artz, Einhorn and Wudka already in the 90s, more systematic and self-consistent. The authors motivate their recipe with a general derivation, that holds for a broad class of UV theories. I find that this methodology can be useful to estimate the expected size of SMEFT corrections to a given process under those specific UV assumptions.

However I strongly disagree with the statement, suggested in several places in the paper, that the EFT power counting is incomplete or even inconsistent, unless the loop suppressions are incorporated into it. (I also disagree with the alternative phrasing that the EFT is unable to give a fully model-independent prediction).

This statement seems to rest on a fundamental misunderstanding: the fact that SMEFT corrections to a given process appear at a certain perturbative order should not be interpreted as a prediction of the size of those effects. For instance, taking the example in Fig. 1: the EFT does NOT imply that diagram (e) gives a subdominant correction compared to (d).
A correct interpretation (assuming that the perturbative expansion holds) is rather that tree-level computations in the EFT capture all possible tree-level BSM effects, one-loop EFT diagrams capture all possible one-loop BSM effects, and so on order by order.

This is actually a quite non-trivial point made by the EFT approach: if one wants to have a model-independent, universal estimate of potential $n$-loop BSM effects, then the EFT calculation can be safely truncated at $n$ loops. The result obtained in this way will be conservative, i.e. it is guaranteed that all $n$-loop BSM contributions will be accounted for, and the matching to a concrete model can only reduce the number of relevant operators.
Reversing this line of reasoning by noting that "the tree-level terms are unable to reproduce the leading [1-loop] corrections of the model" (page 7) simply amounts to a misuse of the EFT.

I really want to stress that loop suppression factors arising in the matching stem from a UV assumption and, from this point of view, they can be considered on the same footing as symmetries of the UV sector, that also act to suppress or forbid certain operator structures.
In fact, the whole reasoning laid out in Section 2 could be repeated identically replacing loop suppressions with some symmetry suppressions. However, one certainly cannot require the power counting to account for all possible symmetry patterns: they can be imposed on top of the EFT expansion on a case by case basis.

I believe that the methodology in Section 3 is valuable and can have a broad applicability, but it has to be made clear that this is not a power counting statement, but rather a size estimate that only holds for specific UV assumptions.
I can only recommend the manuscript for publication if the text is substantially reorganized in this direction.

Requested changes

The only note I make on the physics results (for now) is perhaps a technicality, but I believe that the discussion in Sec. 3.3 assumes that the BSM model:

  1. is weakly interacting in all couplings
  2. does not contain super-renormalizable interactions (i.e. of dimension 3)

If the second assumption is omitted, it is possible to construct models where the naive tree/loop classification does not hold. Some examples were discussed e.g. in 1305.0017 and 1711.10391. In addition, the authors statement that "it is immaterial whether there exist nonrenormalizable interactions of $F$ and $B$ suppressed by scales parametrically still larger than $\Lambda$" only holds if condition 2. is met.

---

## Round 1 · Referee Report · Anonymous (Referee 2) · 2022-9-2

Strengths

The article is pedagogical and easy to follow. It summarises many of the ideas known to those working on higher-order corrections in SMEFT, although not all.

Weaknesses

The article focusses on only the canonical dimension and loop order.

In reality, I don't believe this provides a sufficient organisation of SMEFT terms (which there are clearly many when the full flavour structure of the theory is considered) to practically try and describe limited data.

I believe the authors should also at least comment on this.

Report

The authors present a procedure for arranging corrections in an EFT (focussing on Standard Model Effective Field Theory): this counting includes both loop orders as well as canonical dimension of the operator coefficients.

While these ideas are known to those specifically working on higher-order corrections in the SMEFT (or certainly they are quickly made aware of them, if not), to my knowledge there is not a clear presentation of these ideas with worked examples in the literature. I therefore find this work to be of general use to the community: those working on higher-order corrections, and in addition the groups which are now performing global fits to data (where assumptions on which SMEFT contributions to be included in the fit can have a large impact).

The authors comment towards the end of the manuscript that “Variations of the counting scheme we propose can, of course, be constructed…”. Do the authors have in mind here an alternative counting in terms of loops and canonical dimensions? Or, is this statement hinting at some additional source of organisation, perhaps related to CKM/Yuakwa suppression of interactions (which which may be present under the assumption of minimal flavour violation of the UV theory, or some other symmetry)?

While not a main consideration of the presented work, I believe the authors should at least comment on this. Such considerations are relevant when applying the SMEFT to describe a finite set of data with limited precision.

---

## Round 2 · Referee Report · Anonymous (Referee 1) · 2023-4-28

Report

I thank the authors for addressing my remarks. Clearly we are disagreeing on the meaning of their results.

My point is that the "loop counting" they propose is not an intrinsic property of the EFT (ie not a power counting statement), but the realization of an assumption about the UV model the EFT will be matched to.
Most importantly, it is not a consistency requirement for the EFT as the paper suggests.

The first two examples they mention in their itemize do not represent counterarguments to this statement, because they refer to expanding observables in terms of loops of the theory of interest.
what they claim in their manuscript (and example iii) is rather that the parameters of an EFT should be classified according to a loop expansion of the amplitudes in another theory, the UV completion. Under certain UV assumptions this correspondence certainly exist, but it is not a property of the EFT itself.

Concerning the comparison to other symmetries, consider for instance flavor symmetries. Assuming Minimal Flavor Violation is equivalent to choosing an EFT power counting where flavor violating operators that scale like 2 Yukawa insertions are considered of higher order compared to terms scaling with 1 Yukawa etc. such that the Lagrangian can be truncated at a given order in the Yukawa expansion.
This setup is indeed a power counting choice because it comes from imposing symmetries on the EFT lagrangian itself, and not on a hypothetical UV model (although the two things will be related somehow). When MFV is imposed, it returns nicely consistent results where the suppression pattern is manifest and automatically preserved. Nevertheless, such an assumption is clearly not required for the self-consistency of the EFT. It is an arbitrary choice that one can make, or not.

I do not understand's the authors' comment on the examples in 1305.0017 and 1711.10391.
there is no explicit assumption about the coupling strengths in those papers. the matching expressions descend from simple tree level matching in the presence of super-renormalizable interactions and are valid independent of whether the UV models are strongly or weakly interacting.
these examples represent cases where the loop suppression pattern predicted by loop counting is not respected, which precisely makes the point that loop counting is a legitimate but arbitrary choice on how to order EFT parameters.
  • validity: -
  • significance: -
  • originality: -
  • clarity: -
  • formatting: -
  • grammar: -

Author:  Gerhard Buchalla  on 2023-06-02  [id 3705]

(in reply to Report 1 on 2023-04-28)

We thank the first referee for his feedback on our response.
There may be a disagreement in terminology as to whether
an organizing principle qualifies as power counting or not.
We still believe that our interpretation is justified,
in line with other cases of power counting, and useful in practice.
Let us clarify our point of view by summarizing our main arguments.

We assume a weakly coupled QFT as the new physics at a scale $\Lambda\gg v$.
(If the assumptions were changed, also the power counting would need to be modified.)
This assumption is generic, it covers a very large class of QFT models, while
leaving the heavy-particle content and symmetries unspecified.
In this scenario, any amplitude in the full theory can be expanded in a double expansion:
in the number of loops and in powers of $v^2/\Lambda^2$.
In a low-energy approximation this double expansion has to be reflected in the EFT power counting.
This is the main point of our proposal to include loop counting into the power counting of SMEFT.

Let us try to clarify this further by addressing the objections raised.

a) By consistency in this context we mean that the EFT has to reproduce all the terms at a given order
in the double expansion. We do not imply e.g. that the set of all dimension-6 operators (single insertion)
would not be closed under renormalization, or anything similar.

We have included a clarifying sentence in the Introduction,
at the end of the paragraph "We will argue that ... in perturbation theory.":

“We are not suggesting that SMEFT with a power counting based on canonical dimensions
(with order one operator coefficients) is inconsistent as an EFT under perturbative renormalization.
Rather, we point out that SMEFT organized in such a way would fail to match
a large class of weakly coupled UV models and, in any case, would still call for a reasoning to assign
a power-counting size to the coefficients.”

b) The fact that the loop counting of the UV model is reflected in the EFT power counting doesn't seem
so unusual to us: similarly, the existence of a heavy scale $\Lambda$ is a property of the UV model
and it clearly governs the EFT power counting via the expansion in $v^2/\Lambda^2$.

c) Concerning the examples in 1305.0017 and 1711.10391:
We agree that there is no explicit assumption about the coupling strengths in these papers.
However, either case, weak or strong coupling, may be considered. The power counting
will be different, but consistent with our rules in both cases.
This is also exemplified by a statement we quote from 1711.10391 (Conclusions) and
which is fully in line with our point of view:

"It is interesting that all operators in the Warsaw basis, except for the ones involving
three field strength tensors, are generated in our tree-level integration. This would naively
seem to contradict the arguments in ref. [62], which, up to the presence of L1, share
our assumptions. In fact there is no contradiction since, as we have shown, tree level
contributions to operators that are classified as “loop generated” in [62] only arise due
to non-renormalizable, dimension-five operators in our SM extension, which can only be
generated in turn at the loop level in any weakly-coupled renormalizable UV completion of
that theory. .... However, we have included these operators
in our dictionary because they could be unsuppressed in strongly-coupled completions."

---

## Round 2 · Author Response

Dear Editors,

we thank both referees for reviewing the paper and for their comments.

We first address the remarks made in the second report, which raise the following questions:

{\it 1. The authors comment towards the end of the manuscript that “Variations of the counting scheme we propose can, of course, be constructed…”. Do the authors have in mind here an alternative counting in terms of loops and canonical dimensions? Or, is this statement hinting at some additional source of organisation, perhaps related to CKM/Yukawa suppression of interactions (which may be present under the assumption of minimal flavour violation of the UV theory, or some other symmetry)? }

In fact, both of these aspects may be relevant. Concerning the counting of loops and dimensions, an alternative to SMEFT is the Higgs EFT, organized primarily by chiral counting (loop orders), and thus with a different power counting than SMEFT. Concerning additional sources of organization, we agree that further assumptions can be useful to organize the hierarchies in the flavour sector. However, these issues do not affect (flavour-conserving) electroweak physics, which are our main concern here. A systematic EFT treatment of flavour violation would be beyond the scope of our article.

{\it 2. While not a main consideration of the presented work, I believe the authors should at least comment on this. Such considerations are relevant when applying the SMEFT to describe a finite set of data with limited precision. }

We have included a clarifying statement at the end of the Conclusions (Sec. 5), after the last sentence "Variations of the counting scheme ... explicitly specified.":

"For example, the SMEFT may be replaced by HEFT, which follows a different power counting. In addition, further assumptions, such as minimal flavor violation, can be used to organize the flavor sector, which however is a separate topic and beyond the scope of our paper."

We also thank the first referee for his positive comments on our methodology. However, we do not agree with his additional arguments.

The referee acknowledges the usefulness of our approach by writing e.g.

{\it The methodology they suggest goes in the direction of making the well-known tree/loop operator classification, initially proposed by Arzt, Einhorn and Wudka already in the 90s, more systematic and self-consistent. ... I find that this methodology can be useful to estimate the expected size of SMEFT corrections to a given process under those specific UV assumptions.}

It is therefore somewhat surprising for us that he nevertheless objects to the consequences of this idea, which we have spelled out in our paper.

The main objection of referee 1 is that he questions the interpretation of loop counting as a power counting in EFTs. We disagree with this objection. Our view is fully consistent with common practice in quantum field theory. We mention three examples:

\begin{itemize} \item[i)] Even in a conventional, renormalizable and perturbative QFT (such as QED) the loop counting in the computation of amplitudes plays the role of a power counting: it organizes the perturbative approximation and dictates, which contributions need to be included at any loop order. \item[ii)] In the context of (nonrenormalizable) EFTs, the chiral perturbation theory of pions in QCD is a clear example. This EFT is organized by chiral counting, which is equivalent to a counting of loop orders. \item[iii)] Concerning SMEFT itself, we may consider a scenario where the standard model at the electroweak scale $E\sim v$ is extended by weakly coupled new physics at a scale $\Lambda\gg v$. Any amplitude, in the full theory, can be written as a double expansion in $E^2/\Lambda^2$ and loop factors $1/16\pi^2$. It is apparent that the EFT description of the new physics (SMEFT) has to reflect this double expansion as well. This is what we discuss in the present paper. \end{itemize}

We also disagree with the referee’s view that loop suppressions and symmetries of the UV sector should be considered on the same footing. One might argue that the class of weakly coupled UV models is much larger than the class of models with any specific UV symmetry. More specifically, e.g. the smallness of the electron mass compared to the top-quark mass (presumably due to some underlying flavor symmetry) will justify corresponding approximations in practice, which however are not essential to the EFT computation per se.

We next address the more specific points raised by the referee. He writes:

{\it If one wants to have a model-independent, universal estimate of potential $n$-loop BSM effects, then the EFT calculation can be safely truncated at $n$ loops.}

It is of course true that all possible $n$-loop BSM effects are captured by a full $n$-loop EFT calculation. However, and this is exactly the point of our paper, such an approach would be redundant/overcomplete, since some of the $n$-loop effects are in reality higher loop effects as our toy-model calculation clearly demonstrates. While the approach suggested by the referee is guaranteed to be exhaustive, it introduces superfluous parameters, which make it harder to perform a fit of those parameters.

Finally, the referee remarks

{\it ... I believe that the discussion in Sec. 3.3 assumes that the BSM model: 1. is weakly interacting in all couplings, 2. does not contain super-renormalizable interactions (i.e. of dimension 3). -- If the second assumption is omitted, it is possible to construct models where the naive tree/loop classification does not hold. Some examples were discussed e.g. in 1305.0017 and 1711.10391.}

Item~1 corresponds to what we state (in line with Arzt et al., our ref. [31]). The example in 1305.007 (App.) is no contradiction: The operators ${\cal O}=H^\dagger H X^{\mu\nu}X_{\mu\nu}$ are suppressed by $1/m^2_\sigma$ only, since strong coupling has been assumed (this would incidentally correspond to HEFT). For weak coupling, the $\sigma X^{\mu\nu}X_{\mu\nu}$ interaction would be loop suppressed, leading to a loop suppression for ${\cal O}$ as well. Similar comments apply to 1711.10391.

As we have re-emphasized in our comments above, the suggestion to include loop counting into the power counting of a SMEFT of weakly-coupled new physics is an essential part of our paper. Therefore we see no reason for a substantial reorganization of our presentation.

After having addressed the objections raised in report~1 and included the suggestions of report~2, we hope that our submission can now be accepted for publication in SciPost.

Sincerely yours,

G. Buchalla, G. Heinrich, Ch. M\"uller-Salditt, F. Pandler

---

## Round 2 · List of Changes

In response to report 2, we have included a clarifying statement at the end of the
Conclusions (Sec. 5), after the last sentence
”Variations of the counting scheme ... explicitly specified.”:

”For example, the SMEFT may be replaced by HEFT, which follows a different power counting.
In addition, further assumptions, such as minimal flavor violation, can be used to organize the flavor sector, which however is a separate topic and beyond the scope of our paper.”

---

## Editorial Decision

published